# Coil-globule transitions drive discontinuous volume conserving deformation in locally restrained gels

Tetsuya Yamamoto[1], Yuichi Masubuchi[1] & Masao Doi[2]

The equilibrium volume of a thermoresponsive polymer gel changes dramatically across a temperature due to the coil–globule transitions of the polymers. When cofacially oriented nanosheets are embedded in such a gel, the composite gel deforms at the temperature, without changing the volume, and the response time is considerably shorter. We here theoretically predict that the deformation of the composite gel results from the fact that the nanosheets restrain the deformation of some polymers, while other polymers deform relatively freely. The unrestrained polymers collapse due to the coil–globule transitions and this generates the solvent flows to the restrained regions. The response time of this process is rather fast because solvent molecules travel only by the distance of the size of a nanosheet, instead of permeating out to the external solution. This concept may provide insight in the physics of composite gels and the design of thermoresponsive gels of fast response.

[1] Department of Materials Physics, Nagoya University, Furo-cho, Chikusa-ku, Nagoya 464-8603, Japan. [2] International Research Center for Soft Matter Physics and its Applications, Beihang University, 238 Middle Road of North Fourth Ring Road, Beijing, 100191, China. Correspondence and requests for materials should be addressed to T.Y. (email: tyamamoto@nuap.nagoya-u.ac.jp)

A polymer gel is composed of a polymer network, which is swollen in a solvent and changes its volume by taking in (or squeezing out) solvent molecules[1–3]. Gels of specific polymers (called thermoresponsive polymers) show the volume phase transition, in which the volume of the gel at the thermodynamic equilibrium changes by the factor of tens or more across a temperature[4–6]. Thermoresponsive polymer gels have potential applications to soft actuators[7,8], resizable colloids[9–11], drug delivery systems[7,12], etc. The volume phase transition is driven by the coil–globule transition, in which polymers changes from swollen random coil to collapsed globule in a small window of temperature[13,14]. The dynamics of the volume phase transition is very slow, typically in the order of hours (for a gel of ~1 mm size)[15–17]. The dynamics is accelerated for gels with heterogeneous structures[18,19], dangling bonds[20], amphiphilic conetwork[21], and mobile crosslinks[22,23]. However, the response time of the volume phase transition is not shorter than the order of minutes (for a gel of ~1 mm size)[21], because to change the volume of the gel, solvent molecules must travel by the length scale of the size of the gel through the fine mesh of the polymer network[1–3].

Poly(N-isopropylacrylamide) (PNIPA) gels show the volume phase transition at a temperature, above which these gels collapse. In recent experiments, Aida and coworkers prepared a composite gel, in which solid nanosheets are immobilized in a PNIPA gel[24,25]. These nanosheets are cofacially oriented and aligned in a stack of layers, which are parallel to the nanosheets. Above the transition temperature, the composite gel shows elongation in the normal to the layers and contraction in the parallel to the nanosheets. The volume of the gel does not change during the deformation and the response time of the deformation is in the order of seconds for the gel of ~1 mm size[25]. These features are in contrast to the volume phase transition, with which gels change their volume with a long response time. Understanding the physical mechanisms involved in the volume-conserving deformation may open a new avenue of researches of thermoresponsive gels that respond in a reasonably short time.

The coil–globule transition of PNIPA is driven by the fact that the attractive monomer–monomer interactions dominate the monomer–solvent interactions and the mixing entropy above the transition temperature due to the cooperative dehydration. At a first glance, one may think that the volume-conserving deformation is not driven by the coil–globule transitions because the volume fraction of monomers does not increase when the volume is constant. However, the nanosheets in the composite gel locally restrain the deformation of polymers around these nanosheets, while other polymers deform relatively freely; polymers at the restrained and unrestrained regions may show different conformational changes at the temperature of the coil–globule transition.

Here we theoretically analyze the deformation of the composite gel by taking into account the fact that the gel is locally restrained by the nanosheets. Our theory predicts that the composite gel swells in the normal to the layer and shrinks in the lateral direction above the transition temperature. The direction of the deformation is indeed in agreement with experiments[25]. The deformation is driven by the fact that the polymers in the unrestrained regions collapse in the lateral direction due to the coil–globule transitions and this generates the solvent flow from the unrestrained region to the restrained region, extending the polymers in both of the regions in the normal direction. The volume of the gel does not change during the deformation because solvent molecules do not flow out from the gel. With this mechanism, solvent molecules travel only by the length scale of the size of a nanosheet and thus the response time of the deformation is much shorter than the volume phase transition. The stretching ratio of the gel changes discontinuously at the transition temperature, analogous to the first-order phase transition,

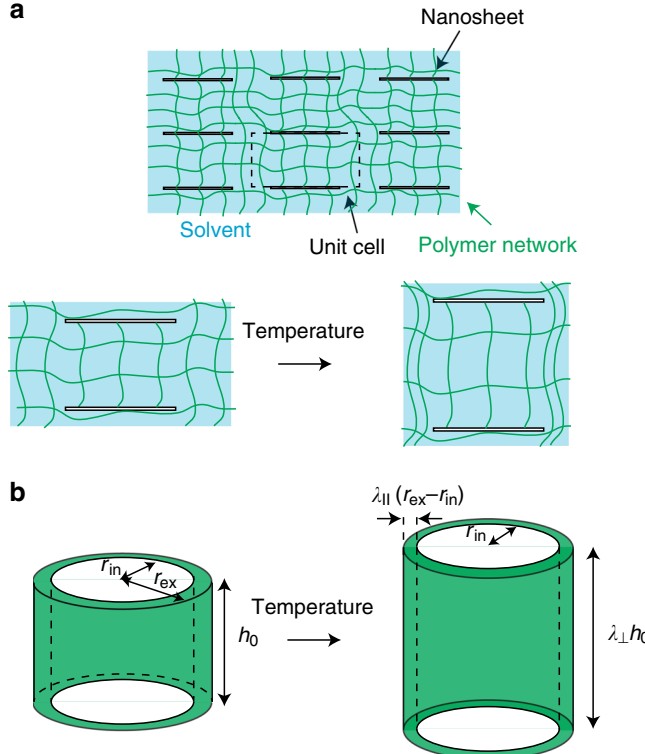

**Fig. 1** A model of a composite gel. **a** The side view of a composite gel, in which solid nanosheets (shown by white rectangles) are immobilized by a polymer network (green lines) in a solvent (light blue background). The nanosheets are cofacially oriented and are aligned periodically. **b** The composite gel is a treated as an assembly of unit cells, which are composed of the central region of radius $r_{in}$, sandwiched by the two registered nanosheets, and the surrounding region of thickness $r_{ex} - r_{in}$. Each unit cell is a cylinder of a radius $r_{ex}$ and the height $h_0$. The thickness $r_{ex} - r_{in}$ of the surrounding region and the distance $h_0$ between the nanosheets are smaller than the radius $r_{in}$ of the nanosheets. The central region can deform in the direction normal to the nanosheets (with an extension ratio $\lambda_\perp$), but cannot deform in the lateral direction. The surrounding region can deform freely in the radial direction (with an extension ratio $\lambda_\parallel$) and can deform in the normal direction with the same extension ratio as the central region. Our theory treats a timescale in which solvent molecules do not permeate out from the gel and thus the volume of each unit cell is constant

even for cases in which the coil–globule transition of the polymers is continuous. Once the mechanism is understood, it may be applied to the design of a thermoresponsive composite gel. The concept of designing the deformations of a composite gel based on the solvent flow between restrained and unrestrained compartments is relatively generic, but it is very different from the conventional designs of composite gels, which respond to the temperature by taking in (or squeezing out) the solvent from the external solution[26,27]. Experimentally testing our predictions may thus open a new avenue of the applications and designs of composite thermoresponsive polymer gels that respond in a reasonably short time.

## Results

**Composite gel model**. We treat a composite gel in which solid nanosheets are immobilized in the polymer network (see Fig. 1a and Methods). The solid nanosheets are cofacially oriented and are aligned in a stack of layers which are parallel to the nanosheets, analogous to the experiments by Aida and coworkers[24,25]. We use a simple model of a polymer gel with which the elastic energy of the polymer network, the mixing entropy, and the interaction energy

are taken into account[2,3] (see Methods and Supplementary Notes 1 and 2 for details). With this model, the coil–globule transition of the polymers of the network is continuous (and thus the gel does not show the volume phase transition when the nanosheets are removed[2,15]). Moreover, there are no long-range interactions (such as electrostatic interactions) between the nanosheets. The magnitudes of the attractive monomer–monomer interactions, relative to the monomer–solvent interactions, are represented by a dimensionless interaction parameter $\chi$[14]. The thermodynamic state of the gel depends on the temperature via the interaction parameter $\chi$. We thus analyze the deformation of the gel when one changes the interaction parameter $\chi$.

For an ideal case in which solid nanosheets are aligned periodically in the layers (Fig. 1), the composite gel is treated as an assembly of unit cells, where each unit cell is composed of the central cylindrical region, sandwiched by two registered nanosheets, and the surrounding region (Fig. 1). We treat cases in which the distance $h_0$ between the two nanosheets and the thickness $r_{ex}-r_{in}$ of the surrounding region are both much smaller than the radius $r_{in}$ of the nanosheets. The solid nanosheets are strongly adhered to polymers in the network and thus restrain the deformation of the polymer network. The central region can deform in the direction normal to the nanosheets (with an extension ratio $\lambda_\perp$), but cannot in the lateral direction; the lateral deformation is suppressed by the elastic energy with respect to the shear deformation of the polymer network for cases in which the aspect ratio $r_{in}/h_0$ is large, see also Supplementary Figs. 1 and 2 and Supplementary Note 4. The surrounding region can deform freely in the radial direction (with an extension ratio $\lambda_\parallel$) and it can deform in the normal direction with the same extension ratio $\lambda_\perp$ as the central region, see Fig. 1b.

The theory of gel dynamics predicts that the timescale with which solvent molecules permeate in a polymer network is proportional to the square of the relevant length scale[2,3]. In a typical experiment, solvent molecules travel through a unit cell (of ~1 μm size) in ~0.1 s and permeate out from the gel (of ~1 mm size) in ~1 day (with a typical value of the diffusion constant of synthetic gels, ~1×10$^{-7}$ cm$^2$ s$^{-1}$, see ref. [1]). We here treat a timescale in which solvent molecules flow between the central and surrounding regions, but do not permeate out from the gel. With this timescale, the volume of each unit cell is constant, see Eq. (12).

**Discontinuous deformation**. We analyze the deformation of the composite gel as a function of the interaction parameter $\chi$. For relatively small values of the parameter $\chi$, the free energy has only one minimum at $\lambda_\perp = 1$, which corresponds to the undeformed state (Fig. 2a). The free energy has a new minimum at $\lambda_\perp > 1$ (which we call the deformed state) when the parameter $\chi$ is larger than a threshold value $\chi_{sp1}$ (Fig. 2b). The free energy at the deformed state decreases, relative to the free energy of the undeformed state, with increasing the parameter $\chi$ and becomes equal to the free energy of the undeformed state at $\chi = \chi_{tr}$. The undeformed state becomes unstable when the interaction parameter $\chi$ is larger than another threshold value $\chi_{sp2}$. This situation is analogous to the first-order phase transition[2]. In general, the composite gel shows a discontinuous deformation at a value of the interaction parameter $\chi$ between $\chi_{sp1}$ and $\chi_{sp2}$. When the parameter $\chi$ is changed very slowly, the extension ratio $\lambda_\perp$ changes discontinuously at the interaction parameter $\chi = \chi_{tr}$, see the vertical broken curves in Fig. 3.

The volume-conserving deformation of the composite gel is driven by the coil–globule transition, where the polymer network shrinks greatly to decrease the free energy for a large interaction parameter $\chi$ (>1/2). Because of the volume conservation of the

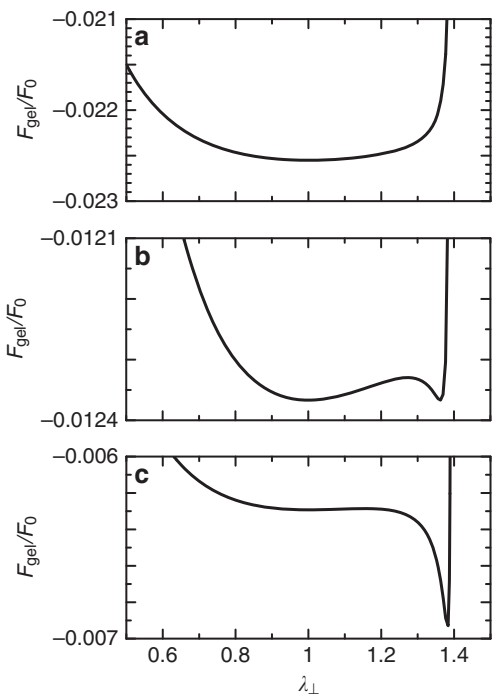

**Fig. 2** Free energy $F_{gel}$ per a unit cell vs the extension ratio $\lambda_\perp$. The free energy $F_{gel}$ per a unit cell (rescaled by $F_0 (\equiv k_B T \pi r_{in}^2 h_0/v_0)$) is shown as a function of the extension ratio $\lambda_\perp$ in the normal of the nanosheets for several values of the interaction parameter $\chi$; **a** $\chi = 0.3$, **b** 0.6132, and **c** 0.8. The values of parameters that are used for the calculations are $s$ (=($r_{ex} - r_{in})/r_{in}$) = 0.2 and $g_0 = 1.0 \times 10^{-3}$ (see Fig. 1 and Eq. (9) for the definition)

unit cell, when the surrounding region collapses in the radial direction to decrease the free energy, the entire unit cell extends in the normal direction. The latter process increases the elastic energy of the polymer network in the central region. This elastic energy plays a role in the energy barrier between the undeformed and deformed states, making the deformation discontinuous (see Fig. 2b and Supplementary Fig. 3). Our theory predicts that the unit cell extends in the normal direction in agreement with experiments[24,25]. This results from the fact that with such a deformation, the normal extension ratio $\lambda_\perp$ (and thus the elastic energy in the central region) is relatively small because the initial volume of the surrounding region is smaller than the initial volume of the central region.

For large values of the interaction parameter $\chi$, the extension ratio $\lambda_\perp$ in the normal direction has an asymptotic form

$$\lambda_\perp = 1 + 2s(1 - \phi_{eq}) \tag{1}$$

(see Supplementary Note 3 for the derivation). $s$ (=($r_{ex} - r_{in})/r_{in}$) is the ratio of the volumes of the central and surrounding regions (see also Eq. (13)). $\phi_{eq}$ is the volume fraction of the polymer network when the composite gel is prepared (see also Eq. (8) and Supplementary Fig. 4). The volume fraction $\phi_{eq}$ is a function of the rescaled shear modulus $g_0$ of the polymer network (which is defined by the ratio of the scale of the elastic energy of the polymer network to the mixing free energy, see Eq. (9)). The rescaled shear modulus $g_0$ increases with increasing the number of crosslinks per unit volume[13]. The extension ratio of the composite gel in the lateral direction has the form $\lambda_\perp^{-1/2}$ due to the conservation of the volume of the unit cell. Our theory predicts that the extension ratio $\lambda_\perp$ (in the deformed state) increases with increasing the volume ratio $s$ (see Fig. 3a and also Eq. (1)) and it decreases with increasing

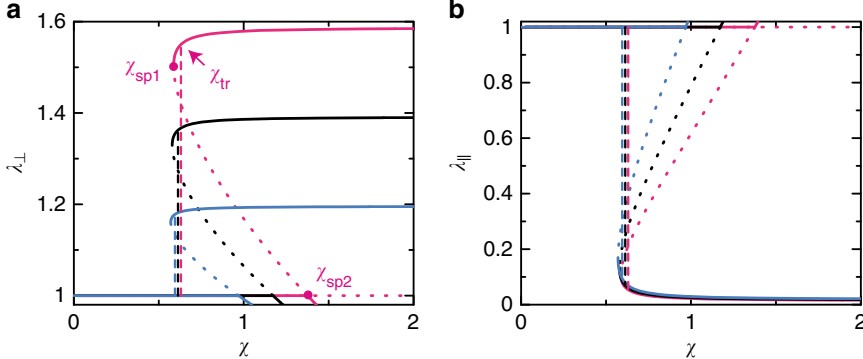

**Fig. 3** The extension ratios vs the interaction parameter $\chi$. The extension ratios, $\lambda_\perp$ (**a**) and $\lambda_\parallel$ (**b**), of a unit cell are shown as functions of the interaction parameter $\chi$ (see Fig. 1 for the definition of $\lambda_\perp$ and $\lambda_\parallel$). We used $(r_{ex} - r_{in})/r_{in} = 0.1$ (cyan), 0.2 (black), and 0.3 (magenta) for the calculations (see Fig. 1 for the definition of the radii $r_{ex}$ and $r_{in}$). The stable solutions are shown by solid curves and the unstable solutions are shown by dotted curves. The global minimum of the free energy changes between the two stable solutions at a threshold interaction parameter $\chi_{tr}$, indicated by broken curves. The value of the rescaled shear modulus $g_0$ (defined by Eq. (9)) is fixed to $1.0 \times 10^{-3}$

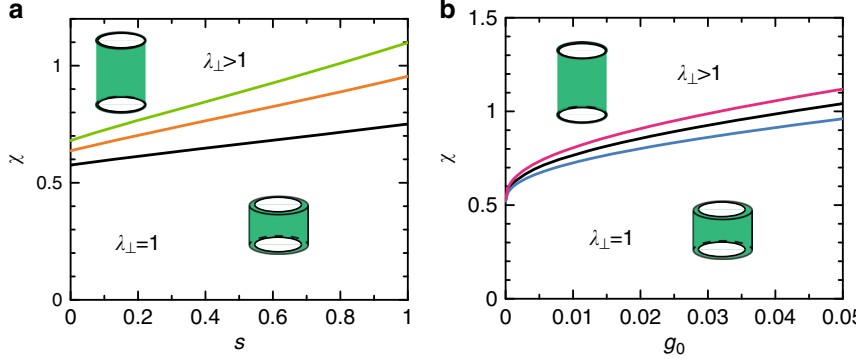

**Fig. 4** Phase diagram of the composite gel. **a** The phase diagrams of a composite gel are shown as functions of the volume ratio $s$ ($\equiv (r_{ex} - r_{in})/r_{in}$) (see Fig. 1 for the definition of $r_{ex}$ and $r_{in}$) for $g_0 = 1.0 \times 10^{-3}$ (black), $5.0 \times 10^{-3}$ (orange), and $1.0 \times 10^{-2}$ (light green), where $g_0$ is the rescaled shear modulus (defined by Eq. (9)). **b** The phase diagram is shown as a function of the rescaled shear modulus $g_0$ for $s = 0.1$ (cyan), 0.2 (black), and 0.3 (magenta)

the rescaled shear modulus $g_0$ of the polymer network (Supplementary Fig. 5). This prediction may be experimentally accessible by measuring the extension ratio $\lambda_\perp$ (or the extension ratio in the lateral direction) as a function of the radius $r_{in}$ of the nanosheets, the lateral distance $2(r_{ex} - r_{in})$ between the nanosheets, and/or the number of cross-links per unit volume.

The response time of the deformation has an approximate form

$$\tau_{gel} = \frac{3}{16} \frac{s}{1 + 2s} \frac{1}{\chi - \chi_{sp2}} \frac{r_{in}^2}{D_{gel}} \tag{2}$$

for $\chi > \chi_{sp2}$ (see Supplementary Note 5 for the derivation). $D_{gel}$ is the diffusion constant with which solvent molecules permeate through the polymer network (defined by Eq. (17)). Equation (2) predicts that the response timescales as $r_{in}^2 s/(1 + 2s)$ with the radius $r_{in}$ of the nanosheets and does not depend on the size of the gel. At the moment when the interaction parameter is changed to $\chi(>\chi_{sp2})$, the osmotic pressure is uniform in the entire region of the gel. A small deformation, driven by the thermal fluctuations, develops the osmotic pressure difference between the central and surrounding regions of a unit cell. This generates the flow of solvent molecules from the surrounding region to the central region. With this mechanism, the solvent molecules travel only by the distance $r_{in}$ during the deformation. This fact is responsible for the fast response of the composite gel. The response time $\tau_{gel}$ decreases with increasing the interaction parameter $\chi$ (Supplementary Fig. 6). These predictions may be experimentally accessible.

**Phase diagram.** When the interaction parameter $\chi$ is changed very slowly, the composite gel shows the discontinuous deformation at a value $\chi_{tr}$. The interaction parameter $\chi_{tr}$ at the transition is a function of the volume ratio $s$ and the rescaled shear modulus $g_0$ of the polymer network. The interaction parameter $\chi_{tr}$ at the transition increases monotonically with increasing the volume ratio $s$ (Fig. 4a). This is because the extension ratio $\lambda_\perp$ of the deformed state increases with increasing the ratio $s$ and it increases the elastic free energy due to the extension in the normal direction (Fig. 3a). The interaction parameter $\chi_{tr}$ increases monotonically with increasing the rescaled shear modulus $g_0$ of the polymer network (Fig. 4b). This is because the elastic energy of the polymer network suppresses the deformation of the composite gel. The interaction parameter $\chi_{tr}$ at the transition has an asymptotic form

$$\chi_{tr} = \frac{1}{2} + \frac{1}{\sqrt{3} 2^{2/5}} (1 + 6s + 4s^2)^{1/2} g_0^{3/10} \tag{3}$$

for cases in which the rescaled shear modulus $g_0$ is small. Equation (3) may be useful to quantitatively test our theory.

When the interaction parameter $\chi$ is changed rapidly, an undeformed gel changes to the deformed state at the second threshold value $\chi_{sp2}$ and a deformed gel changes to the undeformed state at the first threshold value $\chi_{sp1}$ (Fig. 4). These threshold values increase with increasing the volume ratio $s$ and with increasing the rescaled shear modulus $g_0$, analogous to the transition value $\chi_{tr}$ (see Supplementary Figs. 7 and 8 and Supplementary Discussion). Our theory predicts the asymptotic

forms of the threshold values (see Supplementary Eqs. (32) and (38)). The predictions on the interaction parameter $\chi_{tr}$ at the transition (and the threshold values $\chi_{sp1}$ and $\chi_{sp2}$) are probably specific to cases in which the composite gel is composed of polymers that show continuous coil–globule transitions, such as polyacrylamide gels in acetone–water mixture[28] (see also Discussion section).

## Discussion

Our theory predicts that the composite gel shows discontinuous deformation by changing the temperature. The deformation is driven by the fact that the surrounding regions locally collapse to minimize the free energy and thus solvent molecules flow from the surrounding regions to the central regions. The volume of the composite gel is constant during the deformation and the response time is relatively short because solvent molecules travel only by the distance $r_{in}$, corresponding to the radius of the nanosheet. These features are in contrast to the volume phase transition of gels, with which the volume of the gels changes discontinuously and the response time is very long[4–6]. Recent researches showed that a bilayer of the thin films of composite gels is bent or twisted by changing the temperature because the two films change their volume with different ratio[26,27], analogous to bimetal stripes[29]. The latter mechanism is very different from the mechanism of the deformation of the composite gel with layered nanosheets. With our mechanism, the solvent does not permeate through the entire length of the gel and thus the response time is very fast even for a bulk gel.

Our theory predicts that the composite gel shows the discontinuous deformation even for cases in which the coil–globule transition of the polymers is continuous. The discontinuity of the deformation results from the fact that the volume of the unit cell is constant in the relevant timescale. This implies that the discontinuity of the coil–globule transition of PNIPA is not essential for the composite gel to drive the discontinuous deformation. This prediction may be accessible by using the combinations of polymers and solvent (such as polyacrylamide gels swollen in acetone–water mixture[28]) with which the polymers shrink continuously with decreasing the temperature. For cases in which the coil–globule transitions are discontinuous, the polymer gel may show phase separation even without embedded nanosheets and the composite gel made from the polymers thus may show the discontinuous deformations at the temperature range, where coil and globular states are both stable.

Aida and coworkers argued that the discontinuous deformation of the composite gel is driven by the modulation of the electrostatic interactions between solid nanosheets due to the hydration/dehydration of PNIPA polymers at the temperature of the coil–globule transition[25]. This mechanism is only effective for cases in which the dielectric constant changes significantly by the hydration/dehydration and the concentration of mobile ions is relatively small so that the electrostatic interactions are significant. Our theory predicts that the composite gel shows the discontinuous deformation even for cases in which the electrostatic interactions are fully screened by mobile ions. It is of interest to experimentally measure the extension ratio at the transition by changing the concentration of salt impurity. Our theory predicts that the fast response of the composite gel results form the fact that solvent molecules travel only by the distance $r_{in}$. This is true also for cases in which the deformation is driven by the electrostatic interactions.

Our theory provides a couple of experimentally accessible predictions. First, the extension ratio $\lambda_\perp$ in the normal direction increases with increasing the ratio $s$ of the volume of the central and surrounding regions (Fig. 3) and it decreases with increasing

the (rescaled) shear modulus $g_0$ of the polymer network (Supplementary Fig. 5). Equation (1) is a quantitative prediction that can be tested experimentally. This may also be checked via the extension ratio of the gel in the lateral direction, where this extension ratio is $\lambda_\perp^{-1/2}$ due to the volume conservation. The volume ratio $s$ can be changed by changing the size of the nanosheets (or, in some cases, by changing the number of nanosheets per unit volume) and the rescaled shear modulus $g_0$ may be changed by changing the number of crosslinking per unit volume. The extension ratio $\lambda_\perp$ decreases with decreasing the aspect ratio $r_{in}/h_0$, but only slightly (Supplementary Fig. 2a). Second, the response time $\tau_{res}$ of the deformation scales as $sr_{in}^2/(1 + 2s)$ to the radius $r_{in}$ of the nanosheets and it does not depend on the system size (Eq. (2)). Third, for cases in which the polymers in the network show continuous coil–globule transitions, the interaction parameter $\chi_{tr}$ at the transition increases with increasing the rescaled shear modulus $g_0$ of the polymer network and/or the volume ratio $s$ (see Fig. 4 and Eq. (3)).

Although our theory is relatively generic, this theory is ideally tested by experiments that satisfy the following conditions: First, the solid nanosheets are strongly attached to the polymer network, preferably covalently bonded to the polymers (see for example, ref. [24]). In some cases, this condition may be dispensable; the polymer network in the surrounding region is not likely to slide into the central region because the entire region of the gel is connected by the polymer network and the elastic energy of the polymer network suppresses the lateral deformation of the central region (see also Supplementary Note 6). Second, the nanosheets are cofacially oriented and are aligned in a stack of layers (preferably, equally spaced also in each layer), analogous to the composite gels prepared by Aida and coworkers[24,25]. In general, there is a finite variance in the distributions of the size, orientations, and alignment of the nanosheets. Even for cases in which the variance is significant, the timescale of the deformation is still in the order of $\tau_{gel}$ as long as the lateral and normal distances between the nanosheets are smaller than their radius $r_{in}$. In the latter case, the composite gel may deform continuously with increasing the interaction parameter $\chi$. When the orientations of the nanosheets are completely disordered, the deformation of the composite gel, which is presumably isotropic deformation, is prohibited by the conservation of the volume of the gel. Third, the thickness, $r_{ex} - r_{in}$, of the surrounding region (or a half of the lateral distance between the nanosheets) and the normal distance $h_0$ between the two nanosheets are both shorter than the radius $r_{in}$ of the nanosheet. These conditions ensure that the nanosheets locally restrain the deformation of the gel and produce restrained regions and relatively free regions. Fourth, the deformation is measured in a timescale of the order of the response time $\tau_{gel}$, in which solvent molecules travel by the distance of the radius of the nanosheet. In a very long timescale, solvent molecules flow out from the gel and change the volume of the gel, eventually reaching the equilibrium state[3,15]. The process towards the equilibrium indeed corresponds to the volume phase transition, where the lateral contraction (or extension) of the central regions of the unit cells is still restrained by the nanosheets. With a typical value of the diffusion constant of synthetic gels, $D_{gel} \sim 1 \times 10^{-7}$ cm$^2$ s$^{-1}$[1], the timescale of the volume change is ~1 day (for a gel of ~ 1 mm size) and the response time $\tau_{gel}$ is ~0.1 s (for nanosheets of ~1 μm radius). The two timescales are well separated and thus there is a relatively long time period in which our theory is effective. One can also control the above two timescales by designing the radius of the nanosheets, the size of the gel, and the pore size of the polymer network. Fifth, in the swelling process, the gel is swollen in a good solvent (where the interaction parameter is smaller than 1/2). The jump of the extension ratio $\lambda_\perp$ at the threshold of the discontinuous

deformation decreases with increasing the interaction parameter of the solvent used in the swelling process and, eventually, the deformation becomes continuous, see Supplementary Fig. 9.

In conclusion, the discontinuous volume-conserving deformation of the composite gel is driven by the flow of the solvent between the surrounding region and the central region. The local restraint of the gel and the coil–globule transition are enough to drive the solvent flow. Once the mechanism of the deformation is understood, it may be applied to design composite gels. The concept of producing two types of regions with different extent of restraint may be useful to design composite gels, which respond to the temperature in a reasonably short time. Our approach is relatively generic and it is straightforward to extend our theory to cases in which the solid nanosheets show long-range interactions, such as electrostatic repulsion, by including these contributions to the free energy (Methods). Aida and coworkers prepared a composite gel of L-shape to demonstrate that the composite gel shows uni-directional procession when one changes the temperature back and forth, switching between the deformed and undeformed states[24,25]. It is of interest to make a theory that guides the design of such a gel in an extension of our theory. The researches of the bilayer films of composite gel[26,27] are partly motivated by the fact that the actuation mechanism of some plants[30–33] resemble the deformation of these films. Conversely, our theory may provide an option to analyze the actuation mechanism of plants which do not operate with the bilayer mechanism and it may be relevant for cases in which the actuation speed is relatively fast and the volume of the actuation unit is approximately constant. We anticipate that critical experimental tests on our theory open a new avenue of designs and applications of composite gels.

## Methods

**Composite gel**. We treat a composite gel, which is prepared by swelling a polymer network in an athermal solvent (in which the interaction parameter $\chi$ is zero, see Eq. (6) and the discussion below) to the equilibrium and then by embedding solid nanosheets. The solid nanosheets are cofacially oriented and are aligned periodically (Fig. 1). We assume that the solid nanosheets are strongly adhered to the polymer network and thus restrain the deformation of the network. In these cases, the composite gel is treated as an assembly of unit cells, which are composed of the central region, sandwiched by the two registered solid nanosheets, and the surrounding region (Fig. 1). We treat a timescale in which the volume of the gel is constant. Because all the unit cells are equivalent, we analyze the deformations of a unit cell.

**Free energy**. In the reference state (before the network is swollen in a solvent), the position of a material point is represented by using the positional vector $\mathbf{r}_0 = (x, y, z)$ ($x^2 + y^2 \leq r_{ex}^2/\lambda_{eq}^2$ and $0 < z < h_0/\lambda_{eq}$, where $r_{ex}/\lambda_{eq}$ and $h_0/\lambda_{eq}$ is the radius and height of a volume of a polymer network corresponding to a unit cell in the reference state) (Supplementary Fig. 10a). After the deformation, the position of the material point is displaced to $\mathbf{r}(x, y, z)$ (see also Eqs. (7)–(11) below). The deformation of the unit cell is characterized by the metric tensor $g_{\alpha\beta}$ ($\equiv \mathbf{r}_\alpha \cdot \mathbf{r}_\beta$), where $\mathbf{r}_\alpha$ is the derivative of the positional vector $\mathbf{r}(x, y, z)$ with respect to $x_\alpha$. Here and after, the indices $\alpha$ and $\beta$ represent either 1, 2, or 3 and $x_1 = x$, $x_2 = y$, and $x_3 = z$ (Supplementary Methods).

The free energy density of a gel has the form[2,3]

$$F_{gel} = \int dV_0 \left[ f_{ela} + \frac{\phi_0}{\phi} f_{sol} \right], \tag{4}$$

where $f_{ela}$ is the volume density of the elastic energy of the polymer network and $f_{sol}$ is the volume density of the mixing free energy. $\phi_0$ is the volume fraction of the reference state. $\phi$ is the volume fraction after the deformation and has the form $\phi = \phi_0/\sqrt{g}$, where $g$ is the determinant of the metric tensor $g_{\alpha\beta}$. $dV_0$ represents the volume element of the reference state.

The neo-Hookean model treats a polymer network, which is composed of randomly crosslinked Gaussian chains[2]. With this model, the elastic energy density $f_{ela}$ has the form

$$f_{ela} = \frac{1}{2} G_0 (g_{\alpha\alpha} - 3), \tag{5}$$

where $G_0$ is the shear modulus of the polymer network and is proportional to the absolute temperature $T$ for cases in which the elasticity of the polymer network is

entropic[2,13]. $g_{\alpha\alpha}$ is the trace of the metric tensor $g_{\alpha\beta}$. Here and after, we use the Einstein convention, with which repeated subscripts in the same term imply the sum over possible values of the indices. It is straightforward to extend our theory to cases in which the elastic energy has a more elaborate form for quantitative agreement with experiments[34].

The mixing free energy $f_{sol}$ has the form

$$f_{sol} = \frac{k_B T}{v_0} [(1 - \phi)\log(1 - \phi) + \chi\phi(1 - \phi)], \tag{6}$$

where $\chi$ is the dimensionless interaction parameter that represents the magnitudes of the monomer–monomer and solvent–solvent interactions, relative to the monomer–solvent interactions. $k_B$ is the Boltzmann constant and $T$ is the absolute temperature. $v_0$ is the size of a monomer.

With Eqs. (5) and (6), the free energy $F_{gel}$ does not predict the volume phase transition when the nanosheets are removed[2,15].

**Swelling process**. For simplicity, we treat cases in which the polymer network is swollen in an athermal solvent ($\chi = 0$). After the swelling process, the material point at $\mathbf{r}_0 = (x, y, z)$ is displaced to the position

$$\mathbf{r}_s = \lambda_{eq}(x, y, z), \tag{7}$$

where $\lambda_{eq}$ is the swelling ratio. We calculate the free energy of the gel, Eq. (4), by using Eq. (7) in a function of the swelling ratio $\lambda_{eq}$. Taking the derivative of the free energy with respect to the swelling ratio $\lambda_{eq}$ leads to the force balance equation

$$-\frac{G_0}{\lambda_{eq}} + \Pi_{sol}(\phi_{eq}) = 0, \tag{8}$$

where $\Pi_{sol}(\phi) \left( \equiv \phi^2 \frac{\partial}{\partial \phi} \frac{f_{sol}(\phi)}{\phi} \right)$ is the osmotic pressure of the gel (with $\chi = 0$) and $\phi_{eq} (\equiv \phi_0/\lambda_{eq}^3)$ is the volume fraction of the polymer network in the equilibrium (Supplementary Note 1). The volume fraction $\phi_{eq}$ is a function only of the rescaled shear modulus $g_0$ that is defined by the form

$$g_0 = \frac{G_0 v_0}{k_B T \phi_0^{1/3}} \tag{9}$$

(see also Supplementary Fig. 4). Because the shear modulus $G_0$ of the polymer network is proportional to the absolute temperature $T$[2], the rescaled shear modulus $g_0$ does not depend on the absolute temperature $T$.

**Restraint of the gel due to the nanosheets**. We incorporate solid nanosheets in the swollen gel and then increase the interaction parameter $\chi$. The lateral deformation of the central region is suppressed by the elastic energy with respect to the shear deformation for cases in which the aspect ratio $r_{in}/h_0$ is large (Supplementary Note 4 and Supplementary Figs. 1 and 2). We thus assume that the polymer network in the central region does not deform in the lateral direction. The positional vector of a material point in the central region has the form

$$\mathbf{r}_{cen} = \lambda_{eq}(x, y, \lambda_\perp z), \tag{10}$$

where $\lambda_\perp$ is the extension ratio in the normal direction (normal to the nanosheets). The polymer network in the surrounding region can deform in the normal direction with the same extension ratio $\lambda_\perp$ as in the central region and deforms in the radial direction with an extension ratio $\lambda_\parallel$ (Fig. 1). The positional vector of a material point in the surrounding region has the form

$$\mathbf{r}_{sur} = \lambda_{eq}(\lambda_\parallel x_r, y_r, \lambda_\perp z_r), \tag{11}$$

where $x_r$ and $y_r$ are the coordinates in the radial and angular directions, respectively, see Supplementary Fig. 10b. The free energy of the gel is derived as a function of the extension ratios $\lambda_\parallel$ and $\lambda_\perp$ by using Eqs. (10) and (11) (Supplementary Note 2).

In the relevant timescale, solvent molecules may flow between the central and surrounding regions, but they do not flow out from the gel; the volume of the unit cell is constant. The conservation of the unit volume is represented by the form

$$\pi r_{in}^2 h_0 + 2\pi(r_{ex} - r_{in})r_{in} = \pi r_{in}^2 \lambda_\perp h_0 + 2\pi\lambda_\parallel(r_{ex} - r_{in})r_{in}\lambda_\perp h_0. \tag{12}$$

Equation (12) leads to the extension ratio $\lambda_\parallel$ as a function of the extension ratio $\lambda_\perp$ and the ratio of the volumes of the two regions

$$s = \frac{r_{ex} - r_{in}}{r_{in}}, \tag{13}$$

(Supplementary Eq. (21)). By using this relationship, the free energy is represented as a function only of the extension ratio $\lambda_\perp$ (Fig. 2). The extension ratio $\lambda_\perp$ in the stable states is derived by finding the local minima of the free energy with respect to $\lambda_\perp$ (Fig. 3).

**Time evolution equation**. For simplicity, we treat cases in which the thickness $r_{ex} - r_{in}$ of the surrounding region is very small. Solvent molecules thus travel through the central region most of the time and the surrounding region is in the local equilibrium. The time evolution of the deformation of the composite gel is represented by the displacement vector $\mathbf{u}(\mathbf{r}) = (0, 0, \epsilon(t)z_s)$ of the central region, where $\epsilon(t) (=\lambda_\perp - 1)$ is the strain and $z_s$ is the distance from the nanosheet at the bottom of the unit cell (Supplementary Fig. 10c).

The flow field $\mathbf{v}_{sol}(\mathbf{r})$ of the solvent in the central region is derived by using Darcy's law

$$(1 - \phi)(\mathbf{v}_{sol}(\mathbf{r}) - \dot{\mathbf{u}}(\mathbf{r})) = -\kappa \nabla p(\mathbf{r}), \tag{14}$$

where $\phi$ is the volume fraction of the polymer network, $\kappa$ is Darcy's constant, and $p(\mathbf{r})$ is the hydrostatic pressure. Henceforth, the dot above a physical quantity (such as $\dot{\mathbf{u}}(\mathbf{r})$ in Eq. (14)) indicates the time derivative of the quantity. The space is occupied by either solvent molecules or monomers. This is ensured by the condition

$$\nabla \cdot (\phi \dot{\mathbf{u}}(\mathbf{r}) + (1 - \phi)\mathbf{v}_{sol}(\mathbf{r})) = 0. \tag{15}$$

The time evolution equation of the strain has the form

$$\tau_0 \frac{d}{dt}\epsilon(t) = -\frac{1}{K_i + 4G_i/3}\frac{\partial}{\partial \lambda_\perp}\left(\frac{F_{gel}}{\pi r_{in}^2 h_0}\right), \tag{16}$$

where $K_i + 4G_i/3$ is the elastic modulus of the gel ($K_i$ is the osmotic modulus and $G_i$ is the shear modulus) and is defined by the Supplementary Eq. (52). $\tau_0$ is the timescale with which solvent molecules travel by the distance $r_{in}$ and has the form $\tau_0 = r_{in}^2/(8D_{gel})$, see also Equation 146 in ref. [3]. The (effective) diffusion constant $D_{gel}$ of the gel has the form

$$D_{gel} = \kappa(K_i + 4G_i/3), \tag{17}$$

see also refs. [2,3]. Equations (14)–(16) are derived by using the Onsager principle (Supplementary Note 5).

**Response time of the deformation**. We analyze the strain $\epsilon(t)$ as a function of time for cases in which the interaction parameter is changed from zero to $\chi$ at $t = 0$. Equation (16) is analogous to the equation of motion of so-called model A dynamics[35,36]. The right-hand side of Eq. (16) is zero at the undeformed state and thus thermal fluctuations drive the deformation. In the short timescale, the solution of Eq. (16) has an asymptotic form

$$\epsilon(t) = \epsilon_0 e^{t/\tau_{gel}}, \tag{18}$$

which is derived by expanding the right-hand side of Eq. (16) in the power series of the strain $\epsilon(t)$ and omitting the higher order terms. The asymptotic form of the response time $\tau_{gel}$ for small values of the rescaled shear modulus $g_0$ is shown in Eq. (2) (see also Supplementary Eq. (55) for a more general expression). $\epsilon_0$ is the initial strain due to the thermal fluctuations. In the long timescale, the solution of Eq. (16) has an asymptotic form

$$\epsilon(t) = \epsilon_\infty \left(1 - e^{-\alpha_\infty t/\tau_0}\right), \tag{19}$$

where $\epsilon_\infty$ is the strain in the deformed state and the dimensionless parameter $\alpha_\infty$ is defined below the Supplementary Eq. (58) (see also the Supplementary Fig. 11). Equation (19) is derived by expanding the right-hand side of Eq. (16) in the power series of the strain $\epsilon(t) - \epsilon_\infty$ and omitting the higher order terms.

**Data availability**. The Mathematica file (Aidageltheoryver20(revision).nb) used to derive the data that support the findings of this study are available in figshare with the identifier (https://doi.org/10.6084/m9.figshare.6083357).

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

## Acknowledgements

This work was supported by JSPS KAKENHI Grant Numbers JP16H06355, JP18K03558, JP17H01152. M.D. acknowledges the financial support of the National Science Foundation of China (Grant number 21434001).

## Author contributions

T.Y. and M.D. designed the research. M.D. came up with the main idea of the theory. T. Y. made the model and performed the calculations. Y.M. provided important suggestions. All authors wrote and revised the paper.

## Additional information

**Competing interests:** The authors declare no competing interests.

