## [Peer Review File · Nature Communications]

Reviewers' comments:

Reviewer #1 (Remarks to the Author):

In this paper, the authors propose a model for the volume-conserving phase transition of a recently developed thermos-responsive composite gel. It is proposed that the volume-conserving transformation is due to the constraints imposed by the cofacially aligned nano-sheets, which direct the local mass flux of solvent molecules within each unit cell, when the coil-globule transition takes place in the polymer solute. A local equilibrium model is used to compute the anisotropic deformation, and a mixture-theory-like evolution equation is derived from the Onsager principle to estimate the characteristic time of the transformation. While the qualitative picture (direction of expansion) matches with experiments, the model does not provide any checkable quantitative results for verification. Although the proposed mechanism is plausible, and the theory is self-consistent, the reviewer does not think the innovation is significant enough to warrant the publication on Nature Communications.

Technically, the reviewer is not fully convinced by the model for the following reasons:

1) For Darcy-flow-like solvent migration driven by pressure gradient (Eq. S44) (or equivalently gradient of osmotic pressure or chemical potential), where is the pressure gradient initially from? When the gel is brought to a point below the LCST, the distribution of solvent is uniform between the nanosheets and in the outside matrix, the same composition will lead to the same initial osmotic pressure/chemical potential everywhere. In other words, rigid geometric constraints do not add any free energy to the system, and would thus not create any driving force.

2) For directional solvent migration in between the nanosheet, the characteristic time given by r_{in}^2/D still gives a time much longer than that observed when reasonable diffusion constant D is taken.

The reviewer also found the following typos:

p.2, second paragraph, "drag delivery"=>"drug delivery"?

supplementary information p.11, Eq. (S43), a time derivative of the displacement is possibly missing.

Reviewer #2 (Remarks to the Author):

The authors address the deformation mechanism of thermoresponsive hydrogels with integrated co-facially aligned nanosheets through the temperature range where the constituting polymers display a (constrained) coil-globule transition. The understanding of the impact of the co-facially integrated nanosheets on the deformation is sought for using modelling of a representation of the network with the integrated nanosheets. More closely describing the mechanism that can account for the impact of integrating co-facially aligned nanosheets in thermoresponsive hydrogels are highly welcome, and in addition to providing more comprehensive insights, it is also expected to stimulate further experimental realizations of composite nanosheet-hydrogel materials. Although the work appears to be of broad interest for understanding and design of soft materials, and the approach are based on established standards, there is also some concerns, as briefly outlined in the following, that are arising.

The authors are encouraged to more closely discuss the assumption that the changes in deformation profile are within the timeframe where no solvent is expelled from the gel (second paragraph, p. 6). After all, with the unit cell being of the same size as the nanosheets (Fig. 1b), one should have a basis for the actual time.

The prediction that the experimentally observed deformation pattern (reported by others) occur due to solvent flow from the surrounding regions towards the region of the co-facially aligned nanosheets in the cylindrical unit cell considered, appear to focus on the kinetics and not the steady state.

Although the authors are aware that there could be a net change in the polymer volume fraction at longer times (6 lines from bottom, p. 13), the authors are also encouraged to extend the explanation to the new steady state, and offer some more insights on which time scales the various phenomena may occur on.

The possible application of the proposed mechanism as an actuation mechanism in plants (top of page 14) appear premature and should not be included.

A possible typo: Second line, p.6; the statement "...can deform in the normal" should probably read "...can deform in the direction normal"

Citation 11 should be completed

Reviewer #3 (Remarks to the Author):

This manuscript provide a theoretical framework to explain the reversible deformation in locally restrained gels, which is the basis for understanding a wide class of networks such as hydrogels with (thermo)responsive actuation behavior. This is a relatively new field in soft materials (roughly in the last 5 years) drawing from old ideals (thermoreversible gels) and having applications ranging from composite gels with tunable properties to plant biology.

The theory is very elegant and very well presented, with the various assumptions carefully stated. In addition, important links to experiments are made, either by comparing with existing literature, or by suggesting new experiments. The main new idea is that in the restraint gel there are two regions, that of restraint network in contact with the imbedded cofacially oriented nanosheets (constraints) and the unrestraint one, whose difference under coil-globule transition is responsible for solvent flow through the former. The net novel result is the discontinuous deformation (Fig.3) of the (volume preserving) composite gel with fast response time.

Given the importance of hydrogels and responsive materials, the earlier landmark experiments (published in Nature journals but without theoretical explanation) and the potential impact of the work, I believe that it is appropriate for publication in Nature Comm. and that it will have a substantial impact on the field. I have a few comments for the authors to consider:

My main concern pertains to analysis of the restrained gel and the assumption that its central region (far from the nanosheet) cannot deform in the lateral direction. I find this an ad-hoc assumption, albeit not unreasonable. It seems to me that this depends on the gel's modulus (i.e., the crosslink density). For lower modulus (and depending on other factors such as perpendicular deformation), this may not be entirely true and the question is whether that authors agree and what the effect may be (or else what a threshold modulus for this to happen may be). The rest of the analysis follows and is very elegant and insightful, as I mentioned.

The problem definition (selection of unit cell) is very clever and is shown to explain the main effect. Nevertheless, in view of the link to experiments this is a gross approximation. There are several factors such as dispersity of size of nanosheets, not completely parallel sheets, distance between sheets (or cells) not the same – unit cells not equivalent-, making the density of the unrestraint gel heterogeneous, hence its contribution during the coil-globule transition non-uniform. In addition, the (understandable) strong adherence of nanosheets to the polymers although reasonable, makes the problem too specific; it would be interesting to know (or estimate) what the response would be for a given strength with lower energy than covalent, say reversible. That would bring the analysis closer to wider range of potential experimental applications.

Concerning solution thermodynamics, the analysis is based on athermal solvent. It makes sense to have a feeling of the more realistic case of intermediate-quality solvent. Further, in relation to the coil-globule transition, in the case of the unrestraint gel, could this lead to phase separation ? (I see the specific polymers of Aida et al., in particular PNIPA as mentioned by the authors as well).

In the analysis of gel dynamics, isn't the shear modulus involved rather than the bulk? It seems to me that the authors use the bulk (by the way this is all based on Poisson ratio 0.5?).

Concerning the authors' suggestions for experimental tests: The first (electrostatic interactions / continuous coil-globule transition) in p.12 is essentially repeated (second paragraph and third-last paragraph). Checking the normal extension ration is indeed challenging and realistic and here the possible role of gel modulus (see comment above) and lateral extension could be checked as well.

The conditions/restrictions for making the theory relevant are nicely laid out but make the problem more restrictive. I understand that this is a first approach, and still very insightful. But see also previous comments pertaining to nanosheet adhesion and orientation. Finally, to enhance the impact of the work, the ending sentence on potential implications to plan biology could be slightly elaborated. The possibility of considering/discussing the role of adding electrostatic repulsions for closer link to composite gels (and relevance to the recent work of Aida et al., also referring to L-shaped hydrogels) would be useful as well.

Reply to Reviewer #1

Thank you very much for your critical comments. Your comments helped us to make our manuscript more concise. The revised parts are highlighted by bold letters, except for new Supplementary Materials – Supplementary Notes 4 and 6 and Supplementary Figures 1, 2, 5, and 9. Our point-by-point replies follow:

While the qualitative picture (direction of expansion) matches with experiments, the model does not provide any checkable quantitative results for verification.

Our theory indeed predicts an asymptotic form of the extension ratio λ_{\perp} of the composite gel for large value of the interaction parameter χ (when the solvent quality is very poor), see eq. (1). This is a quantitative prediction that can be tested experimentally. To emphasize it, we revised a sentence on the 3rd paragraph, page 13:

Eq. (1) is a quantitative prediction that can be tested experimentally.

For smaller values of the interaction parameter χ , many of the quantities that were predicted by our theory depend on the interaction parameter χ . The treatment of the interaction parameter of thermoresponsive polymers, such as PNIPAM, is indeed tricky. For polymer species that do not show discontinuous coil-globule transitions, one can use the general form

$$\chi = \frac{A}{T} + B$$

where one can find the values of the constants A and B in many polymer handbook. The relationship between the interaction parameter χ and the temperature T for thermoresponsive polymers is shown in refs. (6) and (13) in the main article.

Although the proposed mechanism is plausible and the theory is self-consistent, the reviewer does not think the innovation is significant enough to warrant the publication on Nature Communications.

In this manuscript, we are proposing a concept, with which including nanosheets produces two types of regions – restrained and unrestrained regions, in a gel. The composite gel thus shows a deformation that is different from isotropic deformation,

which is expected from a uniform gel. This deformation is very fast because solvent travels only by the distance in the order of the size of nanosheets. This is a non-trivial mechanism that has not been addressed before (as far as we aware) and would be useful to design a composite gel that deforms in a reasonable time scale. We thus believe that our theory is significant and will motivate experiments and designs of composite gels.

1) For Darcy-flow-like solvent migration driven by pressure gradient (Eq. S44) (or equivalently gradient of osmotic pressure or chemical potential), where is the pressure gradient initially from? When the gel is brought to a point below the LCST, the distribution of solvent is uniform between the nanosheets and in the outside matrix, the same composition will lead to the same initial osmotic pressure/chemical potential everywhere. In other words, rigid geometric constraints do not add any free energy to the system, and would thus not create any driving force.

As you suggested, the nanosheets in the composite gel are not the driving force of the deformation, just produce restrained and unrestrained regions. The driving force of the volume-conserving deformation is the coil-globule transitions of the polymer network. To avoid misunderstanding, we revised the sentences on the 2nd paragraph, page 8:

The volume-conserving deformation of the composite gel is driven by the coil-globule transition, where the polymer network shrinks greatly to decrease the free energy for a large interaction parameter χ ($> 1/2$). Because of the volume conservation of the unit cell, when the surrounding region collapses in the radial direction to decrease the free energy, the entire unit cell extends in the normal direction. The latter process increases the elastic energy of the polymer network in the central region. This elastic energy plays a role in the energy barrier between the undeformed and deformed states, making the deformation discontinuous (see fig. 2b and Supplementary Figure 3). Our theory predicts that the unit cell extends in the normal direction in agreement with experiments^{24,25}. This results from the fact that with such a deformation, the normal extension ratio λ_{\perp} (and thus the elastic energy in the central region) is relatively small because the initial volume of the surrounding region is smaller than the initial volume of the central region.

The dynamics of the volume conserving deformation is shown in Supplementary Note 5. Briefly, the osmotic pressure is uniform in the entire gel at the moment when the temperature is changed. Contracting the surrounding region in the lateral direction drives the expansion of the gel in the normal to the nanosheets because the volume of the gel is constant in the time scale of interest (solvent molecules do not permeate out from the gel). When the interaction parameter χ is large enough, the free energy decrease due to the contraction of the surrounding region dominates the free energy increase due to the expansion of the gel in the normal direction. With thermal fluctuations that contract the surrounding region by small amplitude, the free energy density (or osmotic pressure) of the surrounding region decreases and the free energy density (or osmotic pressure) of the central region increases. The pressure gradient thus forms at the interface between the central and surrounding regions.

To explain the dynamics of solvent molecules, we added sentences on the 1st paragraph, page 10:

At the moment when the interaction parameter is changed to $\chi > \chi_{s2}$, the osmotic pressure is uniform in the entire region of the gel. A small deformation, driven by the thermal fluctuations, develops the osmotic pressure difference between the central and surrounding regions of a unit cell. This generates the flow of solvent molecules from the surrounding region to the central region.

2) For directional solvent migration in between the nanosheet, the characteristic time given by r_{in}^2/D still gives a time much longer than that observed when reasonable diffusion constant D is taken.

With a typical diffusion constant of synthetic gels, $\sim 1 \times 10^{-7}$ cm²/s (see, for example, ref. 1 in the revised manuscript), the response time τ_{gel} of the gel is in the order of ~ 0.1 s for cases in which the radius r_{in} of the nanosheet is in the order of ~ 1 μ m. The response time of the gel, prepared by Aida and coworkers, is approximately 1 s. Our theory shows a reasonable agreement with experiments performed by Aida and coworkers.

To address these time scales, a sentence in the 2nd paragraph, page 6:

In a typical experiment, solvent molecules travel through a unit cell (of 1 μm size) in ~ 0.1 s and permeate out from the gel (of ~ 1 mm size) in ~ 1 day (with a typical value of the diffusion constant of synthetic gels, $\sim 1 \times 10^{-7}$ cm^2/s).

The notified typos were corrected. We also carefully checked our manuscript again to eliminate other typos.

Reply to Reviewer #2

Thank you very much for your constructive comments. Your comments helped us to clarify the time scales that are relevant to the discontinuous volume-conserving deformation and the process towards the equilibrium. The revised parts are highlighted by bold letters, except for new Supplementary Materials – Supplementary Notes 4 and 6 and Supplementary Figures 1, 2, 5, and 9. Our point-by-point replies follow:

1. *The authors are encouraged to more closely discuss the assumption that the changes in deformation profile are within the timeframe where no solvent is expelled from the gel (second paragraph, p. 6). After all, with the unit cell being of the same size as the nanosheets (Fig. 1b), one should have a basis for the actual time.*

With a typical value of the diffusion constant of synthetic gels, $\sim 1 \times 10^{-7}$ cm^2/s , the response time of the volume-conserving deformation is in the order of ~ 0.1 s (for cases in which the size of the nanosheets is in the order of $1 \mu\text{m}$) and the volume of the gel changes in the time scale of ~ 1 day (for cases in which the size of the gel is in the order of 1 mm). These two time scales are thus well separated. To address these time scales, we added a sentence in the 2nd paragraph, page 6:

In a typical experiment, solvent molecules travel through a unit cell (of 1 μm size) in ~ 0.1 s and permeate out from the gel (of ~ 1 mm size) in ~ 1 day (with a typical value of the diffusion constant of synthetic gels, $\sim 1 \times 10^{-7}$ cm^2/s).

2. *The prediction that the experimentally observed deformation pattern (reported by others) occur due to solvent flow from the surrounding regions towards the region of*

the co-facially aligned nanosheets in the cylindrical unit cell considered, appear to focus on the kinetics and not the steady state. Although the authors are aware that there could be a net change in the polymer volume fraction at longer times (6 lines from bottom, p. 13), the authors are also encouraged to extend the explanation to the new steady state, and offer some more insights on which time scales the various phenomena may occur on.

As you see our reply to your comment #1, the time scale of the volume-conserving deformation is well separated from the time scale with which solvent molecules permeate out from the gel. Therefore, it is reasonable to discuss the volume-conserving deformation with our approach, although the composite gel slowly changes its volume to the equilibrium with a long time scale. Moreover, the response time of the volume-conserving deformation is determined by the size of the nanosheets and the time scale of volume change is determined by the size of the gel. One can thus control these time scales by designing the size of the nanosheets and the gel.

The process towards the new steady state (or the equilibrium state) indeed corresponds to the volume phase transition, but the lateral extension (or contraction) of the central regions of the unit cells is still restrained. One can treat this process by using the standard framework of gel dynamics (see for example refs. (3) and (15) in the revised manuscript). To address these points, we added sentences on the 1st paragraph, page 15:

In a very long time scale, solvent molecules flow out from the gel and change the volume of the gel, eventually reaching the equilibrium state^{3,15}. The process towards the equilibrium indeed corresponds to the volume phase transition, where the lateral contraction (or extension) of the central regions of the unit cells is still restrained by the nanosheets. With a typical value of the diffusion constant of synthetic gels, $D_{\text{gel}} \sim 1 \times 10^{-7} \text{ cm}^2/\text{s}$ ¹, the time scale of the volume change is ~ 1 day (for a gel of ~ 1 mm size) and the response time τ_{gel} is ~ 0.1 s (for nanosheets of ~ 1 μm radius). The two time scales are well separated and thus there is a relatively long time period in which our theory is effective. One can also control the above two time scales by designing the radius of the nanosheets, the size of the gel, and the pore size of the polymer network.

3. *The possible application of the proposed mechanism as an actuation mechanism in plants (top of page 14) appear premature and should not be included.*

Thank you very much for this comment. We agree with you that the relationship between our theory and actuation mechanism of plants is premature. We wanted to address our theory also to plant biologists, hoping that our theory is an option to analyze the actuation mechanism of plants that do not operate by the bilayer mechanism. It may be relevant for cases in which the actuation speed is relatively fast and the volume of the actuating unit is (approximately) constant. We had a long discussion among coauthors about whether we should remove the sentence or elaborate it, considering your comment and the comments by the reviewer #3. We finally decided to revise a sentence on the 1st paragraph, page 16, as follows:

Conversely, our theory may provide an option to analyze the actuation mechanism of plants which do not operate with the bilayer mechanism and it may be relevant for cases in which the actuation speed is relatively fast and the volume of the actuation unit is approximately constant.

The notified typos were corrected. We also carefully checked our manuscript again to eliminate other typos.

Reply to Reviewer #3

Thank you very much for your interest to our theory and your constructive comments. Your comments helped us to clarify the rationales behind the assumptions and approximations that are used in our theory and to make the discussion more profound. The revised parts are highlighted by bold letters, except for new Supplementary Materials – Supplementary Notes 4 and 6 and Supplementary Figures 1, 2, 5, and 9. Our point-by-point replies follow:

1. My main concern pertains to analysis of the restrained gel and the assumption that its central region (far from the nanosheet) cannot deform in the lateral direction. I find this an ad-hoc assumption, albeit not unreasonable. It seems to me that this depends on the gel's modulus (i.e., the crosslink density). For lower modulus (and

depending on other factors such as perpendicular deformation), this may not be entirely true and the question is whether that authors agree and what the effect may be (or else what a threshold modulus for this to happen may be). The rest of the analysis follows and is very elegant and insightful, as I mentioned.

In general, as you suggested, the central region shows a lateral deformation, where the amplitude of the deformation is a function of the distance from the nanosheets. We thus performed a new analysis on the amplitudes of the lateral deformation in the central region and the effects of the lateral deformation on the extension ratio λ_{\perp} in the normal direction by using the variational approximation. Briefly, our theory predicts that the amplitudes of the lateral deformation scales as h_0^2/r_{in}^2 and decreases with increasing the aspect ratio of the unit cell r_{in}/h_0 (for cases in which the aspect ratio is large), see Supplementary Figure 1d. The lateral extension in the central region is thus negligible as long as the distance h_0 between the nanosheets is smaller than the radius r_{in} of the nanosheets. These results are summarized in the new supplementary materials – Supplementary note 4 and Supplementary Figures 1 and 2, in the revised manuscript. To clarify the condition with which our approximation is effective, we revised a sentence on the 1st paragraph, page 6, in the main article:

The central region can deform in the direction normal to the nanosheets (with an extension ratio λ_{\perp}), but cannot in the lateral direction; the lateral deformation is suppressed by the elastic energy with respect to the shear deformation of the polymer network for cases in which the aspect ratio r_{in}/h_0 is large, see also Supplementary Figures 1 and 2.

We also added a sentence on the 2nd paragraph, page 19 (in **Methods**), in the main article:

The lateral deformation of the central region is suppressed by the elastic energy with respect to the shear deformation for cases in which the aspect ratio r_{in}/h_0 is large, see Supplementary Note 4 and Supplementary Figures 1 and 2. We thus assume that the polymer network in the central region does not deform in the lateral direction.

2. The problem definition (selection of unit cell) is very clever and is shown to explain the main effect. Nevertheless, in view of the link to experiments this is a gross approximation. There are several factors such as dispersity of size of nanosheets, not completely parallel sheets, distance between sheets (or cells) not the same – unit cells not equivalent-, making the density of the unrestrained gel heterogeneous, hence its contribution during the coil-globule transition non-uniform. In addition, the (understandable) strong adherence of nanosheets to the polymers although reasonable, makes the problem too specific; it would be interesting to know (or estimate) what the response would be for a given strength with lower energy than covalent, say reversible. That would bring the analysis closer to wider range of potential experimental applications.

The distribution of the size, orientation, and spacing of nanosheets

As you suggested, in general, there is a finite variance of the distribution of the size, orientations, and alignment of the nanosheets. To address the effect of the variance, we added sentences on the 2nd paragraph, page 14:

In general, there is a finite variance in the distributions of the size, orientations, and alignment of the nanosheets. Even for cases in which the variance is significant, the time scale of the deformation is still in the order of τ_{gel} as long as the lateral and normal distances between the nanosheets are smaller than their radius r_{in} . In the latter case, the composite gel may deform continuously with increasing the interaction parameter χ . When the orientations of the nanosheets are completely disordered, the deformation of the composite gel, which is presumably isotropic deformation, is prohibited by the conservation of the volume of the gel.

The strong adhesion between the nanosheets and the polymer network

At first, we thought that the strong adherence between the polymer network and the nanosheets is necessary to restrain the deformation of the polymer network in the central region. However, the polymer network in the surrounding region is not likely to slide into the central region because the entire region of the gel is connected by the polymer network and the elastic energy of the polymer network

suppresses the lateral deformation of the central region. To address it, we added sentences on the 2nd paragraph, page 14:

In some cases, this condition may be dispensable; the polymer network in the surrounding region is not likely to slide into the central region because the entire region of the gel is connected by the polymer network and the elastic energy of the polymer network suppresses the lateral deformation of the central region (see also Supplementary Note 6).

3. Concerning solution thermodynamics, the analysis is based on athermal solvent. It makes sense to have a feeling of the more realistic case of intermediate-quality solvent. Further, in relation to the coil-globule transition, in the case of the unrestraint gel, could this lead to phase separation? (I see the specific polymers of Aida et al., in particular PNIPA as mentioned by the authors as well). In the analysis of gel dynamics, isn't the shear modulus involved rather than the bulk? It seems to me that the authors use the bulk (by the way this is all based on Poisson ratio 0.5?).

The quality of solvent during the swelling process

In the previous manuscript, we treated cases in which the gel is swollen in an athermal solvent. However, it is straightforward to take into account the quality of the solvent in an extension of our theory. To address how one can take into account the solvent quality, we added sentences on the 2nd paragraph, page 4 in the Supplementary Information:

For simplicity, we treat cases in which the polymer network is swollen in an athermal solvent ($\chi=0$) in the main article. However, it is straightforward to take into account the quality of the solvent used for the swelling process; it is taken into account by using supplementary equation (S10) with $\chi=\chi_0$ ($\neq 0$) to solve supplementary equation (S9), see Supplementary Figure 9).

We also performed new calculations on the extension ratio for cases in which the gel is swollen in an intermediate solvent. The result is shown in the Supplementary Figure 9. To show the result, we added sentences on the 1st paragraph, page 15:

Fifth, in the swelling process, the gel is swollen in a good solvent (where the interaction parameter is smaller than 1/2). The jump of the extension ratio λ_{\perp} at the threshold of the discontinuous deformation decreases with increasing the interaction parameter of the solvent used in the swelling process and, eventually, the deformation becomes continuous, see Supplementary Figure 9.

Even when a polymer gel is swollen in an intermediate solvent, it does not drive the phase separation in a polymer gel when nanosheets are not embedded.

Phase separation

In general, the swollen and collapsed phases coexist in thermoresponsive gels during the volume phase transition, see for example, refs. 3, 13, and 15. However, in our theory, we use the free energy that does not predict the volume phase transition. Our theory thus predicts that the solvent flow from the surrounding region to the central region is generated even when the gel does not show phase separation. The discontinuity of the coil-globule transitions is thus not essential to the discontinuity of the volume-conserving deformation. We emphasized it in the 3rd paragraph, page 12 in the revised manuscript. However, for cases in which the coil-globule transitions are discontinuous (and thus the polymer gel shows the volume phase transition when the nanosheets are removed), the solvent flows from the surrounding region to the central region may be driven by the phase separation. We thus revised a sentence on the 3rd paragraph, page 12:

For cases in which the coil-globule transitions are discontinuous, the polymer gel may show phase separation even without embedded nanosheets and the composite gel made from the polymers thus may show the discontinuous deformations at the temperature range, where coil and globular states are both stable.

Elastic modulus involved in the gel dynamics

Globally, the volume of the composite gel does not change during the deformation. However, locally, the volume of the surrounding region decreases and the volume of the central region increases during the deformation. The Poisson ratio that

accounts for the local deformation of the gel is thus not 1/2, and both of the osmotic and shear moduli, K_i and G_i , are involved in the gel dynamics, see Supplementary equations (S64)-(S66).

4. Concerning the authors' suggestions for experimental tests: The first (electrostatic interactions / continuous coil-globule transition) in p.12 is essentially repeated (second paragraph and third-last paragraph). Checking the normal extension ration is indeed challenging and realistic and here the possible role of gel modulus (see comment above) and lateral extension could be checked as well.

Repeated sentences

We eliminated the repeated sentence.

Contributions of gel modulus and lateral extension

Motivated by your comments, we showed the extension ratio λ_{\perp} for several values of the rescaled shear modulus g_0 of the polymer network in Supplementary Figure 5. To show the result, we added sentences on the 2nd paragraph, page 9:

Our theory predicts that the extension ratio λ_{\perp} (in the deformed state) increases with increasing the volume ratio s (see fig. 3a and also eq. (5)) and it decreases with increasing the rescaled shear modulus g_0 of the polymer network (see Supplementary Figure 5). This prediction may be experimentally accessible by measuring the extension ratio λ_{\perp} (or the extension ratio in the lateral direction) as a function of the radius r_{in} of the nanosheets, the lateral distance $2(r_{ex} - r_{in})$ between the nanosheets, and/or the number of cross-links per unit volume.

We also revised a sentence on the 3rd paragraph, page 13, where we summarized our predictions:

First, the extension ratio λ_{\perp} in the normal direction increases with increasing the ratio s of the volume of the central and surrounding regions (see fig. 3) and it decreases with increasing the (rescaled) shear modulus g_0 of the polymer network (see Supplementary Figure 5).

Regarding the roles played by the lateral extension, our theory predicts that the lateral deformation in the central region increases with decreasing the aspect ratio r_{in}/h_0 of the unit cell (see Supplementary Figure 1c). The extension ratio thus decreases with decreasing the aspect ratio r_{in}/h_0 (see Supplementary Figure 2). To address it, we added a sentence on the 3rd paragraph, page 13:

The extension ratio λ_{\perp} decreases with decreasing the aspect ratio r_{in}/h_0 , but only slightly (see Supplementary Figure 2a).

In some cases, it may be simpler to measure the extension ratio in the lateral direction. To address the relationship between the lateral extension ratio and the normal extension ratio, we added a sentence on the 2nd paragraph, on page 9:

The extension ratio of the composite gel in the lateral direction has the form $\lambda_{\perp}^{-1/2}$ due to the conservation of the volume of the unit cell.

We also added a sentence on the 3rd paragraph, page 13:

This may also be checked via the extension ratio of the gel in the lateral direction, where this extension ratio is $\lambda_{\perp}^{-1/2}$ due to the volume conservation.

5. Finally, to enhance the impact of the work, the ending sentence on potential implications to plant biology could be slightly elaborated. The possibility of considering/discussing the role of adding electrostatic repulsions for closer link to composite gels (and relevance to the recent work of Aida et al., also referring to L-shaped hydrogels) would be useful as well.

Plant biology

Thank you very much for this comment. We wanted to address our theory also to plant biologists, hoping that our theory is an option to analyze the actuation mechanism of plants that do not operate by the bilayer mechanism. To elaborate it, we revised a sentence on the 1st paragraph, page 16:

Conversely, our theory may provide an option to analyze the actuation mechanism of plants which do not operate with the bilayer mechanism and it may be relevant for cases in which the actuation speed is relatively fast and the volume of the actuation unit is approximately constant.

Extension to nanosheets that show electrostatic repulsion and L-shaped gels

Thank you very much for this comment. To enhance the impact of this work, we elaborated the possibility of extending our theory to cases in which the nanosheets show the electrostatic interactions and to symmetric L-shaped gels that show uni-directional precession on the 2nd sentence, page 15:

Our approach is relatively generic and it is straightforward to extend our theory to cases in which the solid nanosheets show long-range interactions, such as electrostatic repulsion, by including these contributions to the free energy (see Methods). Aida and coworkers prepared a composite gel of L-shape to demonstrate that the composite gel shows uni-directional precession when one changes the temperature back and forth, switching between the deformed and undeformed states^{24,25}. It is of interest to make a theory that guides the design of such a gel in an extension of our theory.

REVIEWERS' COMMENTS:

Reviewer #1 (Remarks to the Author):

The authors seem to have addressed the issues and concerns raised by the reviewer. The paper is recommended for publication.

Reviewer #2 (Remarks to the Author):

In revising the manuscript, the authors have addressed my previous concerns and also issues raised by other referees.

Overall, the revised version is considered as an improved presentation (as compared to the original submission) of a theoretical analysis directed towards understanding of recently discovered phenomena of engineered nanosheet-polymer hydrogels.

Based on the novelty, quality of work and expected interest to a substantial research community, the manuscript is recommended for publication in Nature Communications.

Reviewer #3 (Remarks to the Author):

The authors have thoroughly and convincingly addressed all review comments. Of particular note is that, more than once they have deeply elaborated their replies and, starting from a review question/comments they have extended the discussion beyond the original concern(s) or the reviewer(s). This is highly appreciated. Importantly, it is now evident that the present approach (i) can be tested (in part) experimentally and (ii) can be used to explain / predict a wide range of phenomena.

I conclude that the work is important and timely (as stated in my original review) and will have a substantial impact in the field. Hence, I support its publication in Nat. Comm.

Reviewer #1

Thank you very much for your constructive comments and recommendation. Your comments helped us to clarify our argument about the physics behind the discontinuous deformation.

Reviewer #2

Thank you very much for your constructive comments and recommendation. Your comments helped us to clarify the relevant time scale of the theory.

Reviewer #3

Thank you very much for your constructive and inspiring comments. Your comments helped us to make our discussion more profound and enhance the impact of the work.